# Multi-Perspective Test-Time Prompt Tuning for Global, Local Visuals, and Language

## Abstract

Recent advances in vision-language models (VLMs) have demonstrated significant generalization across a broad range of tasks through prompt learning. However, bridging the distribution shift between training and test data remains a significant challenge. Existing researches utilize multiple augmented views of test samples for zero-shot adaptation. While effective, these approaches focus solely on global visual information, neglecting the local contextual details of test images. Moreover, simplistic, single-form textual descriptions limit the understanding of visual concepts, hindering the transfer performance of classes with similar or complex visual features. In this paper, we propose a **M**ulti-**P**erspective **T**est-Time **P**rompt **T**uning method, **MP-TPT**, building on two key insights: local visual perception and class-specific description augmentation. Specifically, we introduce local visual representations from VLMs during the optimization process to enhance the prompts' ability to perceive local context. On the other hand, we design a data augmentation method at the text feature level that imparts regional visual priors to specific class texts, thereby enriching the class-specific descriptions. Furthermore, we synchronize the multi-view concept during the inference, integrating both local and global visual representations with text features for a deeper understanding of visual concepts. Through extensive experiments across 15 benchmark datasets, we demonstrate the advantages of MP-TPT, particularly achieving a 1% improvement in state-of-the-art TPT accuracy in cross-dataset settings, along with 4.5 times acceleration in inference speed.

## 1 Introduction

Pre-trained vision-language models (VLMs), such as CLIP (Radford et al., 2021), establish strong baselines for prompt engineering (Zhou et al., 2022; Wortsman et al., 2022). These models are trained on large-scale image-text pairs, aligning visual and language modality within a shared embedding space. At inference, users can input a hand-crafted prompt as a query to identify the class with the highest similarity to the test image in a zero-shot manner. However, designing heuristic prompt templates tailored to different domains is both labor-intensive and suboptimal. To further unlock the potential of pre-trained VLMs, recent works (Chen et al., 2023; Fu et al., 2024) propose prompt tuning that replaces hand-crafted prompts as a set of learnable context vectors, enabling automatic construction of prompt templates under supervision of downstream datasets. Nevertheless, the quality of such prompt learning is heavily constrained by the distribution of the training data, leading to distributional biases during testing (Abdul Samadh et al., 2024; Yao et al., 2024). Moreover, this approach relies on high-quality annotated data, which may be scarce and expensive to obtain.

In this context, a new paradigm known as Test-Time Prompt Tuning (TPT) (Shu et al., 2022) has been proposed to mitigate domain shift problem in prompt tuning without the need for task-specific training data. Specifically, TPT optimizes text prompts by enforcing consistency learning through entropy minimization across augmented views. However, we observe that TPT can easily fall into the trap of the global visual information from augmented views, neglecting the detailed concepts of the object. For instance, Figure 1(a) shows that most of the retained augmented views focus on high-confidence classification of prominent features (*e.g.*, the face), while capturing some finer details (*e.g.*, the skin) is rare, leading to overfitting on incorrect classes. DiffTPT (Feng et al.,

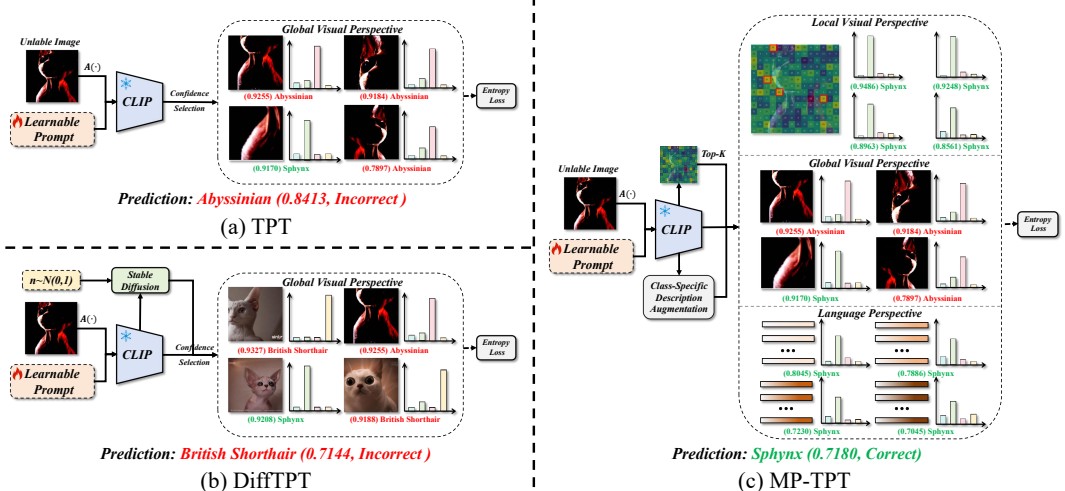

Figure 1: We illustrate the comparison between MP-TPT and the state-of-the-art methods, TPT (Shu et al., 2022) and DiffTPT (Feng et al., 2023), during the tuning phase. Existing methods primarily focus on perceiving augmented views, lacking sensitivity to image context and the text space, which can lead to misfitting. In contrast, our approach leverages the internal knowledge of VLMs to seamlessly integrate perceptions of both "local visual" and "language", resulting in more accurate inference.

2023), externally introduces a diffusion model to enrich the pool of views, but it still struggles to generate critical input details and may even introduce new high-confidence errors, as illustrated in Figure 1(b). This raises an important consideration: the perception of local context in images is a crucial factor for identifying similar or complex classes within TPT. Our core impetus stems is based on the widely accepted consensus that local visual representation of VLMs contain richer contextual information (Chen et al., 2022; Lafon et al., 2024). To clarify, we compute the similarity between text features and local visual features, selecting semantically relevant local regions to participate in the optimization as local views, thereby enhancing TPT's capacity for local context perception.

Image-level augmentations, such as parameter transformations(Shu et al., 2022) and image generation (Feng et al., 2023), introduce additional prior knowledge into the bootstrapping paradigm of TPT, so as to capture the benefits of consistency regularization. Although effective, the single-form nature of class-specific description during the TPT optimization process limits prompts to capturing only image-level augmentation information, hindering the exploration of a broader augmentation space. Recent studies (Tian et al., 2024; Zheng et al., 2024) have shown that generating visual descriptions related to specific class using large language models (LLMs) can serve as a form of text-level augmentation, aligning better with visual concepts. However, LLMs-based methods are not suitable for the TPT setting due to their significant inference overhead. Therefore, we propose a simpler yet effective approach: leveraging local visual features to inject region-specific information into class-specific text features. Specifically, we randomly perturb the cross-modal information between local visual and text features to generate visual prior. These priors serve as a form of data augmentation for the text modality, creating multiple variants of text features that provide rich, class-specific visual descriptions. Furthermore, we extend this concept to the test-time inference phase. In contrast to previous methods (Yoon et al., 2024; Zanella & Ben Ayed, 2024) that focused solely on global visual and text prompts, we leverage local visual representations to enhance text prompts, providing a deeper understanding of class-specific cues.

Overall, it is evident that considering only the diversity of views in the TPT setting is overly simplistic. A more comprehensive approach requires multi-perspective perception, incorporating both local contextual information of the test samples and rich class-specific descriptions. Therefore, in this work, we introduce a multi-perspective optimization method, MP-TPT. As illustrated in Figure 1(c), MP-TPT leverages the inherent local features of VLMs to deepen the understanding of localized visual concepts while generating multiple visual priors to enhance class-specific descriptions. This results in superior visual alignment and significantly enhances the model's adaptability at test time. Additionally, the introduction of multi-perspective views allows for more flexible data

augmentation, thereby enhancing inference efficiency. To sum up, our contributions are as follows: (1) We introduce MP-TPT, a refined method designed for test-time prompt tuning. MP-TPT offers adaptability during both the tuning and inference stages, allowing for efficient integration into existing workflows. (2) MP-TPT uniquely integrates local visual concepts and class-specific descriptions augmentation, marking the first instance in test-time prompt tuning where the focus extends beyond global visual representations. (3) Extensive experiments validate the effectiveness of MP-TPT, significantly enhancing test-time prompt tuning for VLMs.

## 2 RELATED WORK

### 2.1 TEST-TIME ADAPTATIONS

Test-time adaptation (TTA) (Niu et al., 2022; Zhao et al., 2023; Prabhudesai et al., 2023) aims to bridge the distribution gap between training and test data by adapting a pretrained model from the source domain to an unlabeled target domain before making predictions. One popular approach involves minimizing entropy either across batches of test samples (Wang et al., 2021) or multiple views of a single sample (Zhang et al., 2022), which effectively improves test-time accuracy. Our work builds on and extends the discussion of test-time adaptation in VLMs.

### 2.2 PROMPT LEARNING IN VLMS

Prompt learning enhances the adaptability of vision-language models (VLMs) to downstream tasks by introducing learnable text or visual prompts. For instance, CoOp (Zhou et al., 2022) aligns learnable text prompts with task-specific visual knowledge, while VPT (Jia et al., 2022) proposes incorporating visual prompts within Vision Transformers (ViTs) (Dosovitskiy, 2020) to achieve more efficient performance transfer without full fine-tuning. MaPLe (Khattak et al., 2023) builds on this by extending prompt learning to multimodal branches, allowing for the joint learning of deeper prompts. Despite the effectiveness of these methods in transferring VLMs, they heavily depend on high-quality training data and fail to explicitly address distribution shifts at test time. To address this gap, recent work has introduced TTA technology (Shu et al., 2022), which learns text prompts by minimizing the entropy of multiple augmented views of the test sample. Subsequent methods enhance TPT by incorporating external tools such as diffusion models (Feng et al., 2023), and reward models (Zhao et al., 2024), but these approaches incur significant computational overhead, making them unsuitable for test-time settings. More importantly, these approaches focus solely on the diversity of global visual representations, limiting their perceptual scope. To overcome these challenges, our method expands the perceptual capabilities by incorporating local visual concepts and class-specific description augmentation, thereby achieving more effective test-time adaptation.

## 3 METHODOLOGY

In this section, we first introduce the preliminary definitions relevant to this work, followed by a detailed description of the proposed MP-TPT framework. This framework comprises two stages: test-time tuning and test-time inference, which integrate global visual, local visual, and language perspectives to enhance test-time adaption, as illustrated in Figure 2.

### 3.1 PRELIMINARY

**Contrastive Language-Image Pre-training.** The pre-trained CLIP model, denoted as $\theta = \{\mathbf{E_V}, \mathbf{E_T}\}$, consisting of two encoders. The visual encoder $\mathbf{E_V}$ typically utilizes a CNN (He et al., 2016) or ViT (Dosovitskiy, 2020) architecture to project visual inputs into a high-dimensional feature space, while the text encoder $\mathbf{E_T}$ employs a Transformer architecture (Vaswani, 2017) to generate corresponding features from a sequence of word tokens. During the training phase, CLIP performs contrastive loss Chen et al. (2020) on approximately 400 million image-text pairs, aiming to maximize the cosine similarity between visual and language embeddings, thus achieving superior modality alignment. For testing, given an image $\boldsymbol{x}$, the visual encoder extracts a global representation $\boldsymbol{f}^v = \mathbf{E_V}(\boldsymbol{x}) \in \mathbb{R}^d$, where $d$ is the feature dimension. For downstream tasks involving $K$ classes, each class is incorporated into a hard prompt formatted as "`a photo of a {class}`",

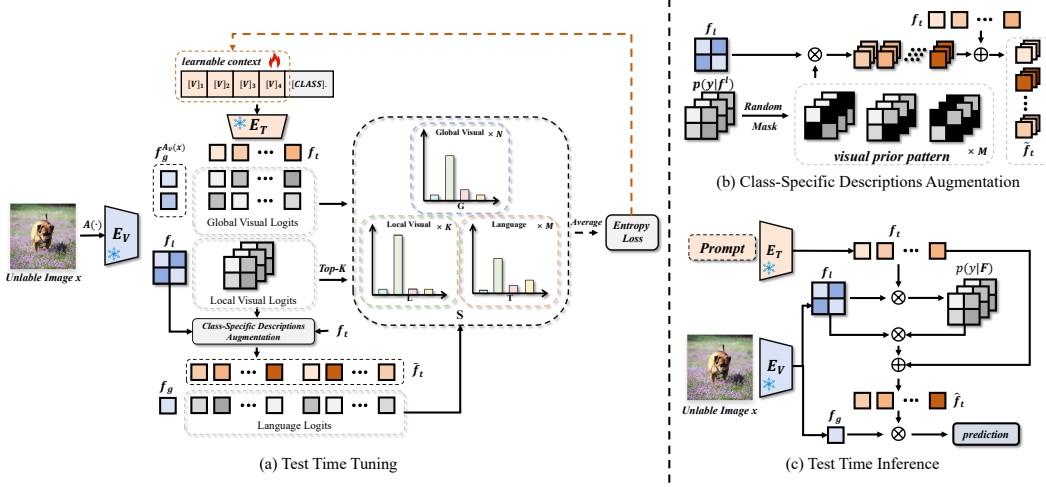

(a) Test Time Tuning

(b) Class-Specific Descriptions Augmentation

(c) Test Time Inference

Figure 2: Overview of our proposed zero-shot image classification method, MP-TPT. (a) Test Time Tuning: We introduce local visual and text level views alongside augmented views to update prompts through entropy minimization. (b) Class-Specific Description Augmentation: Random perturbations of cross-modal information $p(y \mid \boldsymbol{f}^l)$ yield $M$ augmented patterns with regional visual priors, which are injected into original text features. (c) Test Time Inference: Interacting fine-tuned prompts with local visuals generates enriched text features, calculating CLIP similarity with test image global features.

resulting in the text class description matrix $\mathbf{P} \in \mathbb{R}^{K \times l}$, where $l$ is the length of text sequences. The text encoder $\mathbf{E_T}$ encodes $\mathbf{P}$ to produce the text features $\{\boldsymbol{f}_k^t\}_{k=1}^K$, where $\boldsymbol{f}_k^t \in \mathbb{R}^d$ denotes the text feature of the class-specific text input. the prediction probability for image $\boldsymbol{x}$ with respect to class $y_k$ is computed based on the similarity between the visual and text features, expressed as:

$$p(y_k|\boldsymbol{x}) = \frac{\exp\left(\cos\left(\boldsymbol{f}^v, \boldsymbol{f}_k^t\right)/\tau\right)}{\sum_{k=1}^K \exp\left(\cos\left(\boldsymbol{f}^v, \boldsymbol{f}_k^t\right)/\tau\right)}, \tag{1}$$

where $cos(\cdot)$ calculates the cosine similarity between vectors, and $\tau$ is the temperature of the softmax function.

**Test Time Prompt Tuning.** Building on the exceptional performance of CLIP, Test-time prompt tuning introduced by (Shu et al., 2022) aims to leverage the extensive knowledge embedded in CLIP to enhance its generalization capabilities in a zero-shot setting. TPT employs an unsupervised framework to learn a set of prompt vectors $\boldsymbol{V}$ for each test image. As shown in Figure 1(a), TPT consists of three key steps: (1) Data augmentation $\mathcal{A}_v(\boldsymbol{x}_{\text{test}})$ is applied to the single test image, increasing data diversity. (2) The augmented views, along with the prompt $\boldsymbol{V}$, are fed into CLIP to generate corresponding logits, denoted as $p(y \mid \mathcal{A}_v(\boldsymbol{x}_{\text{test}}))$, followed by filtering out high-entropy (low-confidence) logits. (3) The mean entropy of the selected logits is minimized, and the prompt is updated using this information. The detailed process is as follows:

$$\boldsymbol{V}^* = \arg\min_{\boldsymbol{V}} -\sum_{k=1}^K \tilde{p}_{\boldsymbol{V}}\left(y_k \mid x_{\text{test}}\right) \log \tilde{p}_{\boldsymbol{V}}\left(y_k \mid x_{\text{test}}\right), \tag{2}$$

$$\text{where} \quad \tilde{p}_{\boldsymbol{V}} = \frac{1}{\rho N}\sum_{i=1}^N \mathbb{I}\left[\mathbf{H}\left(p_i\right) \leq \tau\right] p_i\left(y \mid \mathcal{A}_v(x_{\text{test}})\right), \tag{3}$$

where $\mathbf{H}(\cdot)$ computes the self-entropy of the prediction probability distribution, and the indicator function $\mathbb{I}\left[\mathbf{H}\left(p_i\right) \leq \tau\right]$ selects $\rho$ percent of the most confident samples based on a cutoff threshold $\tau$.

## 3.2 MP-TPT

In this section, we present our proposed test-time adaptation method, MP-TPT, which is grounded in two key insights: Local Visual Perception and Class-Specific Descriptions Augmentation. As

illustrated in Figure 2(a), we introduce two novel views to enable a broader scope of prompt learning based on these insights. Furthermore, as shown in Figure 2(c), in contrast to most previous approaches that focus solely on inference with global visual information and text prompts, MP-TPT employs a triadic reasoning process involving "global visual," "local visual," and "language", thereby further enhancing inference performance.

### 3.2.1 LOCAL VISUAL PERCEPTION

TPT is built on optimizing the global visual representation based on a set of augmented views. However, alongside data diversity, the perception of image context is equally crucial. This insight motivates us to introduce the local visual representation $\boldsymbol{f}_i^l \in \mathbb{R}^d$, which is obtained by projecting the visual $\tilde{\boldsymbol{f}}_i^l \in \mathbb{R}^D$ of each region $i$ features from the feature map into the text space, as follows:

$$\boldsymbol{f}_i^l = Proj_{v \to t}(\tilde{\boldsymbol{f}}_i^l), \tag{4}$$

where $Proj_{v \to t}(\cdot)$ denotes the projection from visual space to text space, a process inherent in CLIP that does not require additional training. Consequently, we leverage this intrinsic knowledge to obtain rich local context. Subsequently, we establish the relationship between regions and class information based on a set of region indices $I = \{0, 1, 2, \ldots, H \times W - 1\}$, where H and W represent the height and width of the feature map. Analogous to Eq. 1, we compute the similarity between the visual features of each region $i$ and the text features to derive the classification prediction probabilities for each region. The formulation is as follows:

$$p(y_k \mid \boldsymbol{f}^l) = \frac{\exp\left(\cos\left(\boldsymbol{f}^l, \boldsymbol{f}_k^t\right)/\tau\right)}{\sum_{k=1}^{K} \exp\left(\cos\left(\boldsymbol{f}^l, \boldsymbol{f}_k^t\right)/\tau\right)}. \tag{5}$$

$p(y_k \mid \boldsymbol{f}^l) \in \mathbb{R}^{WH \times K}$ encapsulates the strength of association between each region and the class information. Given that class names typically correspond to foreground attributes, it can be reasonably inferred that regions related to the foreground will exhibit high-probability peaks. In contrast, background regions, having a weaker semantic relationship with the class information, tend to display lower probability peaks. Consequently, we select the Top-K regions with the highest prediction probabilities as the set of logits at the local visual level $\boldsymbol{L}$ for optimization, as follows:

$$\boldsymbol{L} = \{p(y \mid \boldsymbol{f}_i^l) : \text{Top-}K(\arg\max_y p(y \mid \boldsymbol{f}_i^l), i \in I)\}. \tag{6}$$

### 3.2.2 CLASS-SPECIFIC DESCRIPTIONS AUGMENTATION

In the MP-TPT framework, the introduction of class-specific descriptions augmentation, as illustrated in Figure 2(b), aims to generate multiple descriptions for specific classes, thereby achieving better alignment between visual information and prompts. Specifically, we apply a random masking operation on cross-modal information obtained from Eq. 5, denoted as $\text{Mask}(p(y_k|\boldsymbol{f}^l)) \in \mathbb{R}^{WH \times K}$. Intuitively, this perturbation is analogous to performing random cropping on the input image, enabling us to capture visual concepts from different regions within the feature space. We regard this as a data augmentation pattern that interacts with the local visual feature $\boldsymbol{f}^l \in \mathbb{R}^{WH \times d}$, generating augmented text features $\{\tilde{\boldsymbol{f}}_i^t\}_{i=1}^M$, where $\tilde{\boldsymbol{f}}_i^t$ represents class-specific descriptions across different regions, and $M$ denotes the number of augmented text features. To preserve the original textual concepts, we also apply a residual operation, detailed as follows:

$$\tilde{\boldsymbol{f}}^t = \alpha \cdot ((\sigma(\text{Mask}(p(y_k|\boldsymbol{f}^l))^{\mathbb{T}} \times \boldsymbol{f}^l) + \boldsymbol{f}^t, \tag{7}$$

where the hyperparameter $\alpha$ controls the extent of the text augmentation, $\sigma(\cdot)$ denotes the softmax function, and $\mathbb{T}$ represents the transpose operation. We compute the similarity between the augmented text features $\tilde{\boldsymbol{f}}^t$ and the global features $\boldsymbol{f}^g$ of the test images to obtain the set of logits at the text level, denoted as $\boldsymbol{T}$. The formulation is as follows:

$$\boldsymbol{T} = \{p(y_k \mid \tilde{\boldsymbol{f}}_i^t) : p(y_k \mid \tilde{\boldsymbol{f}}_i^t), i \in \{0, 1, \ldots, M-1\}\}, \tag{8}$$

$$\text{where} \quad p(y_k \mid \tilde{\boldsymbol{f}}_i^t) = \frac{\exp\left(\cos\left(\boldsymbol{f}^g, (\tilde{\boldsymbol{f}}_i^t)_k\right)/\tau\right)}{\sum_{k=1}^{K} \exp\left(\cos\left(\boldsymbol{f}^g, (\tilde{\boldsymbol{f}}_i^t)_k\right)/\tau\right)}. \tag{9}$$

Table 1: Comparison of MP-TPT in cross-dataset generalization evaluation. Bold indicates the best results, and underlining represents the second-best results.

| | Flowers102 | DTD | OxfordPets | StanfordCars | UCF101 | Caltech101 | Food101 | SUN397 | FGVCAircraft | EuroSAT | Average | Inference Time |
|---|---|---|---|---|---|---|---|---|---|---|---|---|
| CLIP | 67.28 | 44.44 | 88.06 | 65.28 | 65.03 | 92.94 | 83.82 | 62.59 | 23.82 | 41.38 | 63.46 | 0.039 ± 0.001 |
| TPT | 69.31 | 46.99 | 87.38 | 65.99 | 68.01 | 94.00 | 84.73 | 65.43 | 23.07 | 42.81 | 64.77 | 0.583 ± 0.005 |
| C-TPT | 69.71 | 46.16 | 88.23 | 65.43 | 65.40 | 93.43 | 84.61 | 64.45 | 24.42 | 43.28 | 64.33 | 0.583 ± 0.002 |
| MTA | 67.64 | 45.15 | 87.90 | **67.31** | **68.22** | 94.00 | 84.61 | 65.19 | 23.91 | 41.35 | 64.43 | 0.551 ± 0.004 |
| DiffTPT | 70.10 | 47.00 | 88.22 | 67.01 | 66.69 | 92.49 | **87.23** | **65.74** | **25.60** | 43.13 | 65.47 | – |
| MP-TPT-S | **71.05** | **47.81** | **89.02** | 64.92 | 66.77 | 93.96 | 83.86 | 65.10 | 24.21 | 46.71 | 65.34 | **0.131 ± 0.004** |
| MP-TPT-L | 70.12 | 47.28 | 88.83 | 66.29 | 67.54 | **94.01** | 84.60 | 65.32 | 23.34 | 49.23 | **65.66** | 0.584 ± 0.001 |

### 3.2.3 TEST TIME TUNING AND TEST TIME INFERENCE

As outlined above, we obtain a global visual logits set $G$ containing $N$ enhanced views through data augmentation, while the local visual logits set $L$ is derived from local visual perception. Additionally, we generate a text-level logits set $T$ through augmented class-specific descriptions. These sets are then integrated to form a powerful view space $S = \{p_0^s, p_1^s, p_2^s, \ldots, p_{N-1}^s, p_N^s, p_{N+1}^s, \ldots, p_{N+K-1}^s, p_{N+K}^s, p_{N+K+1}^s, \ldots, p_{N+K+M-1}^s\}$. Following this, we update the prompts $V$ using the robust entropy minimization unsupervised paradigm in TPT, as follows:

$$V^* = \arg\min_{V} -\sum_{k=1}^{K} \hat{S}_V(y_k) \log \hat{S}_V(y_k),  \tag{10}$$

$$\text{where} \quad \hat{S}_V = \frac{1}{N+K+M} \sum_{i=0}^{N+K+M-1} p_i^s.  \tag{11}$$

Differing from previous methods that focused solely on aligning prompts with global visuals, we further introduce alignment between prompts and local visuals during the tuning phase. To facilitate this, we propose a dual interaction of text prompts with both global and local visuals during inference, as illustrated in Figure 2(c). Specifically, we utilize the optimized prompts $V^*$ to generate new text features $f^{t*}$, which then interact with local visuals $f^l$ to produce new cross-modal information $p_{V^*}(y_k \mid f^l)$. Subsequently, we perform matrix multiplication between $\sigma(p_{V^*}(y_k \mid f^l))$ and $f^l$ to obtain text features $\hat{f}^{t*}$ enriched with local visual information. By merging $f^{*t}$ and $\hat{f}^{*t}$, we enable a more comprehensive understanding of visual concepts. Ultimately, this leads to the calculation of classification probabilities $p_{V^*}(y_k|x_{\text{test}})$ in conjunction with global features $f^g$, as expressed below:

$$p_{V^*}(y_k|x_{\text{test}}) = \frac{\exp\left(\cos\left(f^g, (f^{t*} + \lambda \cdot \hat{f}^{t*})_k\right)/\tau\right)}{\sum_{k=1}^{K} \exp\left(\cos\left(f^g, (f^{t*} + \lambda \cdot \hat{f}^{t*})_k\right)/\tau\right)},  \tag{12}$$

where, $\lambda$ represents the degree of local visual enhancement introduced.

## 4 EXPERIMENTS

In this section, we describe the tasks and benchmarks used to evaluate our approach, along with the implementation details. Following the standard practices in TPT (Shu et al., 2022), our primary results cover two key aspects of model generalization: Domain Generalization and Cross-Datasets Generalization, as detailed in Sections 4.1and 4.2, respectively. Additionally, Section 4.3 presents ablation studies, analyzing the different network components for test-time tuning, the generality of our insights, and various design choices of our method.

Table 2: Comparison of MP-TPT in domain generalization evaluation. Bold indicates the best results, and underlining represents the second-best results.

| Method | ImageNet | ImageNet-A | ImageNet-V2 | ImageNet-R | ImageNet-Sketch | Average | OOD Average |
|---|---|---|---|---|---|---|---|
| CLIP | 67.30 | 47.14 | 59.90 | 71.20 | 43.00 | 57.71 | 55.31 |
| TPT | 69.70 | 53.67 | 64.30 | 73.90 | 46.40 | 61.59 | 59.57 |
| DiffTPT | **70.30** | **55.68** | **65.10** | 75.00 | 46.80 | 62.28 | 60.52 |
| MP-TPT | 69.00 | 54.44 | 63.28 | **76.92** | **48.04** | **62.34** | **60.67** |
| CoOp | 72.30 | 49.25 | 65.70 | 71.50 | 47.60 | 61.27 | 58.51 |
| TPT+CoOp | 73.30 | 56.88 | 66.60 | 73.80 | 49.40 | 64.00 | 61.67 |
| DiffTPT+CoOp | **75.00** | **58.09** | **66.80** | 73.90 | 49.50 | 64.12 | 61.97 |
| MP-TPT+CoOp | 73.80 | 55.52 | 66.30 | **77.77** | **49.83** | **64.64** | **62.35** |

## 4.1 Cross-Datasets Generalization

Test time adaptation is a key technology for real-world applications, aimed at classifying any category in a zero-shot manner without relying on a training set. Consequently, cross-dataset performance generalization and inference efficiency are critical metrics for TTA methods, which we will analyze in this section.

**Datasets.** We utilize 10 classification datasets that cover a wide range of visual recognition tasks. This includes species of plants or animals (Flowers102 (Nilsback & Zisserman, 2008) and OxfordPets (Parkhi et al., 2012)), transportation (StanfordCars (Krause et al., 2013) and FGVC-Aircraft (Maji et al., 2013)), food (Food101 (Bossard et al., 2014)), satellite (EuroSAT (Helber et al., 2019)), human actions (UCF101 (Soomro, 2012)), texture (DTD (Cimpoi et al., 2014)), scene (SUN397 (Sun et al., 2020)), and general object (Caltech101 (Fei-Fei et al., 2004)).

**Baselines.** To evaluate our proposed method on cross generalization, we compare it against three groups of approaches: (1) TPT (Shu et al., 2022) and its two variants, C-TPT (Yoon et al., 2024) and DiffTPT (Feng et al., 2023), (2) MTA (Zanella & Ben Ayed, 2024), a state-of-the-art training-free TTA method based on augmented views, (3) the classic zero-shot CLIP (Radford et al., 2021) with the default prompt `"a photo of a"`. In this setup, we reproduced all the above baselines, except for DiffTPT, where we directly use the reported results from the original paper due to the time-consuming nature of image generation.

**Implementation Details.** In all experiments, we employ the publicly available CLIP model with the ViT-B/16 (Dosovitskiy, 2020) visual encoder as the backbone. Following the TPT setup, we initialize the prompt with the default hand-crafted phrase `"a photo of a"` and optimize the corresponding 4 tokens based on a single test image. We introduce two versions of MP-TPT: MP-TPT-S, optimized for faster inference, and MP-TPT-L, designed for stronger performance. MP-TPT-S generates $N = 8$ enhanced views via simple parameter transformations, extracts $K = 8$ local views from CLIP, and creates $M = 4$ textual views without filtering any of the views. In contrast, MP-TPT-L produces $N = 64$ enhanced views, $K = 32$ local views, and $M = 32$ textual views, retaining 10% of the views based on a minimum entropy criterion. Unless otherwise specified, we use one-step optimization for prompt tuning during the testing phase, utilizing Adam as the optimizer. The initial learning rate, $\alpha$, and $\lambda$ hyperparameters are set to 0.005, 0.1, and 0.1, respectively.

**Results.** In Table 1, we evaluate the cross-dataset generalization performance of our method. Both variants of our approach demonstrate significant improvements in both speed and accuracy. Notably, MP-TPT-S achieves an average accuracy improvement of 0.57% over TPT using only 8 augmented views, while delivering approximately 4.5 times faster inference (0.583 sec./image *vs.* 0.131 sec./image). On the other hand, the MP-TPT-L variant matches TPT in inference speed while outperforming it by 0.9%, further validating that our multi-view approach introduces no additional computational overhead. In comparison to the state-of-the-art DiffTPT, our method achieves an average accuracy gain of 0.19%. Moreover, DiffTPT incurs substantial time costs due to its significantly higher number of forward passes (128 *vs.* 64), larger optimization steps (4 *vs.* 1), and the time involved in image generation. This indicates that our method is likely over twice as fast in terms of inference. Furthermore, compared to train-free methods, although we employ a single back propagation step, we significantly reduce the number of forward propagation, thereby greatly enhancing inference speed. These results strongly validate the flexibility and powerful generalization capabilities of our multi-perspective perception strategy in TPT.

Table 3: Ablation study on each component. $G$, $L$, and $T$ represent the contributions of data augmentation, local visual perception, and class-specific description augmentation, respectively.

|  | Flowers102 | DTD | Oxford-Pets | Caltech101 | EuroSAT | Average |
|---|---|---|---|---|---|---|
| $G$ | 70.44 | 46.51 | 87.82 | 93.59 | 41.02 | 67.88 |
| $L$ | 68.90 | 46.10 | 87.08 | 92.78 | 47.78 | 68.52 |
| $T$ | 69.18 | 46.04 | 88.77 | 93.06 | 46.86 | 68.78 |
| $G + L$ | 70.16 | 47.22 | 87.90 | 93.31 | 46.54 | 69.02 |
| $G + T$ | 70.32 | 46.22 | 88.63 | 93.59 | 44.79 | 68.71 |
| $L + T$ | 69.27 | 46.81 | 87.46 | 93.02 | 47.60 | 68.83 |
| $G + L + T$ | 71.01 | 46.70 | 88.97 | 93.75 | 46.38 | 69.36 |

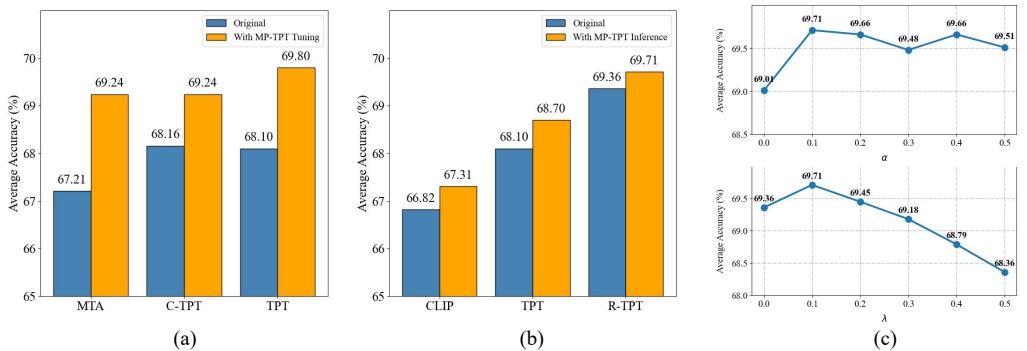

Figure 3: Analysis of the generality of MP-TPT's tuning and inference methods, as well as the effects of different hyperparameter settings. (a) Generality of the tuning process. (b) Generality of the inference process. (c) Ablation study on the hyperparameters $\alpha$ and $\lambda$.

## 4.2 DOMAIN GENERALIZATION

CLIP has been shown to exhibit exceptional robustness to naturally occurring distribution shifts in real-world scenarios. In this section, we follow the setups of previous methods to evaluate the effectiveness of our approach in domain generalization.

**Datasets.** In the Domain Generalization setting, we evaluate our approach on four out-of-distribution (OOD) variants of ImageNet (Deng et al., 2009): ImageNetV2 (Recht et al., 2019), ImageNet-Sketch (Wang et al., 2019), ImageNet-A (Hendrycks et al., 2021b), and ImageNet-R (Hendrycks et al., 2021a).

**Baselines.** We compare MP-TPT with few-shot prompt tuning and test-time prompt tuning methods to validate the effectiveness of our proposed approach. For few-shot prompt tuning, we adopt a standard baseline, CoOp (Zhou et al., 2022), which adjusts the prompt distribution on each downstream dataset. Additionally, we compare MP-TPT with TPT (Shu et al., 2022) and DiffTPT (Feng et al., 2023) to demonstrate its superiority in test-time settings. All comparative results are sourced from (Feng et al., 2023).

**Implementation Details.** We evaluate MP-TPT-L for domain generalization using the same setup as in Section 4.1. Additionally, we initialize the learnable prompts with the pre-trained CoOp weights, which are trained on ImageNet using 16-shot training data per class and 4 learnable prompt tokens, as provided by the official implementation.

**Results.** In Table 2, our MP-TPT demonstrates superior performance compared to both TPT and DiffTPT. Moreover, by applying MP-TPT to the prompts learned by CoOp, we effectively leverage the domain-specific distributional insights from ImageNet, further improving generalization capabilities on OOD data. Notably, our method proves highly effective for domain generalization, achieving significant accuracy gains, particularly on ImageNet-V2, which assesses robustness to co-location, and ImageNet-R, which evaluates robustness across multiple domains.

Table 4: Effect of the number of views across three perspectives. $N, K, M$ represent the number of views for global visual, local visual, and language, respectively.

| N | K | M | Flowers102 | DTD | Oxford-Pets | Caltech101 | EuroSAT | Average |
|---|---|---|---|---|---|---|---|---|
| 4 | 4 | 4 | 70.89 | 47.28 | 88.96 | 93.67 | 47.30 | 69.62 |
| 8 | 8 | 8 | 70.77 | 47.64 | 89.18 | 93.83 | 47.05 | 69.66 |
| 16 | 16 | 16 | 71.34 | 47.93 | 88.63 | 93.63 | 46.65 | 69.64 |
| 8 | 8 | 64 | 69.47 | 46.87 | 88.74 | 93.18 | 47.25 | 69.10 |
| 8 | 64 | 8 | 68.66 | 46.99 | 88.17 | 93.31 | 48.96 | 69.22 |

## 4.3 ABLATION STUDY

In this section, detailed analyses are shown to help understand the superiority of our MP-TPT, including effectiveness of different components, the general applicability of our approach across different test-time adaptation techniques, and analysis of different hyperparameter settings. For simplicity, all ablation experiments are evaluated on five datasets (Caltech101, DTD, Flowers102, Oxford-Pets and EuroSAT).

**Effectiveness of Different Components.** We have conducted a comprehensive set of experiments, as detailed in Table 3, to substantiate the effectiveness of our two principal techniques. To ensure a fair comparison and fully capture the performance of each component, we utilized the same inference setup as TPT. Our results reveal a striking variation in generalization capabilities across different datasets. For instance, in the fine-grained dataset, data augmentation achieved an impressive accuracy, yet it struggled on the more coarse-grained satellite dataset. In contrast, the local view showed remarkable performance in satellite image classification, while the class-specific description augmentation significantly boosted transferability on the Oxford-Pets. These findings underscore the critical importance of multi-view integration, proving that the synergy between diverse perspectives is indispensable for achieving superior generalization in test-time prompt tuning.

**Generality of Our Tuning and Inference Method.** Our approach comprises two crucial phases: tuning and inference. In Figure 3, we demonstrate its generality across various methods. As shown in Figure 3(a), our tuning technique yields substantial improvements in MTA, C-TPT, and TPT. Notably, in the training-free MTA setting, we propose integrating local views into the quality assessment variables and directly incorporating them into the optimization process, resulting in a performance boost of over 2%. Additionally, as depicted in Figure 3(b), we incorporate local features into the inference process for CLIP, TPT, and our MP-TPT. The results show that even on the zero-shot baseline, this inference strategy proves highly effective, underscoring its robustness across diverse scenarios.

**Analysis of Different Hyperparameter Settings.** In Figure 3(c), we analyze two hyperparameters related to local visual information: $\alpha$ and $\lambda$. Specifically, $\alpha$ controls the incorporation of local visual cues in class-specific descriptions augmentation. The results show a marked improvement in performance when visual priors are involved in text descriptions. Additionally, $\lambda$ represents the degree to which local information is embedded in text during the inference stage. We observe that accuracy initially improves as $\lambda$ increases but eventually declines, suggesting that over-reliance on localized context can hinder generalization. In Table 4, we assess three key hyperparameters that define our approach: the number of views for each perspective. Increasing the number of views for local visual and language perspectives individually yields better performance on some datasets but negatively impacts overall generalization. In contrast, simultaneously increasing views across all perspectives results in stable performance.

## 5 CONCLUSION

In this paper, we introduced MV-TPT, a novel method that enhances test-time adaptation (TTA) for vision-language (VLMs), facilitating zero-shot generalization. Our approach improves generalization capabilities by incorporating local visual and language perspectives. Specifically, we leverage inherent local visual representations from VLMs during optimization and design class-specific description augmentations that include visual priors. Extensive experiments demonstrate that MV-TPT achieves competitive performance and plug-and-play capabilities. We believe our insights will significantly benefit the TTA community.

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

.

