# OpenReview forum: "Multi-Perspective Test-Time Prompt Tuning for Global, Local Visuals, and Language"
_ICLR.cc/2025/Conference — ICLR 2025 Conference Withdrawn Submission_

### Official Review · Reviewer_yWFV · 2024-11-01

**Soundness:** 3
**Presentation:** 3
**Contribution:** 2
**Rating:** 5
**Confidence:** 4

**Summary:**

The paper introduces a novel method called Multi-Perspective Test-Time Prompt Tuning (MP-TPT) designed to enhance vision-language models (VLMs) during test time. Unlike prior approaches that focus solely on global visual features, MP-TPT combines global and local visual information with language prompts, offering a comprehensive view during test-time adaptation. The method enhances textual prompts with class-specific descriptions by using local visual information, which allows the model to capture diverse contextual variations. Extensive experiments across multiple benchmarks demonstrate that MP-TPT achieves notable improvements in accuracy and inference speed compared to state-of-the-art methods, particularly in zero-shot and cross-dataset generalization scenarios.

**Strengths:**

1. MPTPT addresses a critical limitation in existing methods that rely solely on global visual features. By incorporating class-specific, region-based prompts, the paper proposes an innovative way to adapt VLMs to unseen data without retraining, which is both effective and practical.
 2. The methodology is rigorous, with extensive experiments on 15 benchmark datasets that demonstrate the model's adaptability and efficiency, especially in zero-shot and cross-dataset settings. Ablation studies add further credibility by detailing each component's contribution.

**Weaknesses:**

1. Limited improvement over global feature methods: results indicate that the performance gains of MP-TPT over other methods focusing on global visual features, such as DiffTPT, are not substantial, which raises questions about the effectiveness of incorporating local visuals.
2. The paper does not sufficiently clarify the interaction between local visual features and text descriptions. A more detailed explanation of how these components integrate during optimization and inference would enhance understanding.
3. While MP-TPT introduces local visual information to improve class-specific descriptions, the paper could benefit from a deeper analysis of how these local augmentations influence specific categories, particularly when handling complex classes.

**Questions:**

1. It would strengthen your claims to include more comprehensive comparisons with a broader range of state-of-the-art methods in your experiments. Highlighting specific scenarios where MP-TPT excels or falls short could provide valuable insights.
2. Can you clarify the specific roles that global, local, and language perspectives play in test-time prompt tuning? In particular, how do local and language perspectives interact, considering their apparent strong coupling.
3. Could you provide more experiment on MPTPT+CoOP/MaPLE or other prompt tuning method in Base-to-Novel Generalization? It will help to prove MPTPT’s effectiveness as a plug-to-play prompt learning mthod.
4. Providing detailed ablation studies that analyze trade-off between speed, accuracy and amount of parameters would enhance the understanding of the practical implications of MP-TPT.

---

### Official Review · Reviewer_Y52q · 2024-11-03

**Soundness:** 2
**Presentation:** 2
**Contribution:** 2
**Rating:** 3
**Confidence:** 4

**Summary:**

This paper proposes utilizing local visual context and class-specific text description augmentation to improve the classification accuracy of the test-time prompt tuning of CLIP model. The local visual representation is obtained by projecting the entire visual feature to the region level and calculating the similarity with text features. The top-K high-similarity region features are selected to produce the class-specific descriptions. The prompts and the global-local visual features are further aligned through a dual interaction during the tuning phase. Experiments show some improvement.

**Strengths:**

The motivation is clear. Existing test prompt tuning methods focus only on global visual feature augmentation, neglecting the importance of local context in images. By introducing fine-grained local visual features and their corresponding text prompt descriptions, the proposed method should contribute to improved test-time prompt tuning results. The paper is easy to understand.

**Weaknesses:**

1. The main weakness of this paper is that the experimental results are marginal. From Table 1, we can see that the best result of the proposed MP-TPT (65.66) is only 0.2% better than the baseline DiffTPT (65.47). Similarly, in Table 2, the MP-TPT method also shows a marginal improvement (less than 0.5%). Did the authors conduct statistical significance tests to verify the effectiveness of the proposed method? These minor differences may also stem from the randomness of the training process. Providing error bars or standard deviations would make the results more convincing. Furthermore, does the method work beyond the CoOp framework, such as on Maple[1] and PromptSRC[2]?

[1] MaPLe: Multi-modal Prompt Learning
[2] Self-regulating Prompts: Foundational Model Adaptation without Forgetting

**Questions:**

1. In L107, how can the method enhance inference efficiency when it requires multi-perspective views, which will obviously increase computational and storage costs? Additionally, Table 1 shows that MP-TPT-S has a lower inference time than TPT. What are the different experimental settings between these two methods, and is the comparison fair? Could the authors provide a more detailed analysis of computational complexity and memory usage?

2. The description in Section 3.2.3 is difficult to understand. What is the difference between test time tuning and test time inference? How to generate $\boldsymbol{f}^{t *}$ and $\hat{\boldsymbol{f}}^{t *}$? Additionally, Figure 2c is confusing; how is Eq. 12 applied in Figure 2c, e.g, where is the $\lambda$ in Figure 2c?

---

### Official Review · Reviewer_7h8V · 2024-11-04

**Soundness:** 2
**Presentation:** 1
**Contribution:** 2
**Rating:** 3
**Confidence:** 5

**Summary:**

This paper studies the topic of test-time prompt tuning (TPT), and propose MP-TPT. MP-TPT introduces local patch features as additional visual augmentations, which may be crucial for classification. Additionally, it leverages local visual features to enhance text feature descriptions. Extensive experiments demonstrate the effectiveness of the proposed method.

**Strengths:**

1. The motivation to use local features is intuitive, as they may contain important details that can enhance model performance.

2. The paper conducts extensive experiments, including comparisons of MP-TPT on two representative benchmarks and ablation studies.

**Weaknesses:**

1. **Limited novelty and contribution**: The concept of using local features to enrich image features and image counterparts to enhance text features has already been proposed in [1]. The primary difference in this paper is the implementation of this idea in a test-time prompt tuning scenario. Surprisingly, [1] is not cited or discussed in this paper.

2. **Clarity and organization**: The paper is difficult to follow due to disorganized writing, confusing formulas, and figures and tables that are not self-contained. This impacts its suitability for ICLR acceptance. I list some below:

   1. Line 225: The term "local visual representation" is unclear. Is this referring to CLIP patch features? This needs clarification.
   2. Line 253: Why are classification probabilities referred to as "cross-modal information"? Is it simply because they use features from two modalities? What specific information do they contain?
   3. In Equation (7), The resulting shape is $\mathbb{R}^{W H \times d}$. How are $M$ augmented features derived?
   4. In Equation (7), There are 5 left brackets and 3 right brackets, making the expression difficult to understand.
   5. In Table 1, how is the inference time calculated? Are the times in seconds? Different datasets with varying classes should have different inference speeds. The table should be self-contained.
   6. Multiple definitions of $K$: In Line 161, $K$ is defined as the number of classes, while in Line 244 and Equation (6), $K$ is the number of selected regions.
   7. Undefined terms: $\boldsymbol{f}^t$ in Equation (7) is not defined. Is it a set or a concatenation of $\boldsymbol{f}^t_i$?
   8. The definition of a set in Equation (8) is incorrect. The part “$p\left(y_k \mid \tilde{\boldsymbol{f}}_i^t\right)$” after the colon should be removed.

3. **Experimental issues**:

   1. The claims in Line 28 are misleading. MP-TPT did not achieve a 1% improvement over TPT and 4.5 times faster simultaneously. These are achieved by different methods, MP-TPT-L and MP-TPT-S.

   2. Some highly relevant works, such as [2] and [3], are missing from Tables 1 and 2. The performance of MP-TPT is significantly lower compared to these methods. More discussion is needed.

      | Methods         | Cross-dataset | Domain Generalization |
      | --------------- | ------------- | --------------------- |
      | PromptAlign [2] | 66.92         | 63.55                 |
      | TDA [3]         | 67.53         | 63.89                 |
      | MP-TPT-L        | 65.66         | 62.35                 |

   3. The ablation study is unconvincing. Why are results provided only on 5 datasets? The proposed methods can lead to performance degradation in many cases, such as in the Flowers102 and Caltech101 datasets. The average performance gain seems to stem from the EuroSAT dataset, which only contains 10 classes and is sensitive.

4. **Effectiveness of design**: The use of random masks on local features as a proxy for random cropping is questionable. I explored this idea in test-time prompt tuning tasks a year ago and found it ineffective, raising concerns about its effectiveness in MP-TPT.

5. **Lack of error bar analysis**: The paper does not include an error bar analysis, which is an important aspect of experimental evaluation.

[1] Task-Oriented Multi-Modal Mutual Learning for Vision-Language Models. ICCV 2023.

[2] Align Your Prompts: Test-Time Prompting with Distribution Alignment for Zero-Shot Generalization. NeurIPS 2023.

[3] Efficient Test-Time Adaptation of Vision-Language Models. CVPR 2024.

**Questions:**

Intuitively, local patch features do not align with text features and therefore cannot be directly utilized, as studied in [1]. Could the authors provide more discussion or visualizations to illustrate this aspect?

[1] A Closer Look at the Explainability of Contrastive Language-Image Pre-training.

---

### Note · Authors · 2024-11-21

I have read and agree with the venue's withdrawal policy on behalf of myself and my co-authors.